# Beyond Hemostasis: Platelet Innate Immune Interactions and Thromboinflammation

**DOI:** 10.3390/ijms23073868

**Published:** 2022-03-31

**Authors:** Jonathan Mandel, Martina Casari, Maria Stepanyan, Alexey Martyanov, Carsten Deppermann

**Affiliations:** 1Center for Thrombosis and Hemostasis, University Medical Center of the Johannes Gutenberg-University, 55131 Mainz, Germany; jonathan.mandel@uni-mainz.de (J.M.); macasari@uni-mainz.de (M.C.); mstepany@uni-mainz.de (M.S.); 2Center For Theoretical Problems of Physico-Chemical Pharmacology, 109029 Moscow, Russia; malexmart95@gmail.com; 3Physics Faculty, Lomonosov Moscow State University, 119991 Moscow, Russia; 4Dmitriy Rogachev National Medical Research Center of Pediatric Hematology, Oncology Immunology Ministry of Healthcare of Russian Federation, 117198 Moscow, Russia; 5N.M. Emanuel Institute of Biochemical Physics RAS (IBCP RAS), 119334 Moscow, Russia

**Keywords:** platelets, hemostasis, thrombosis, neutrophils, monocytes, macrophages, inflammation, NETs, COVID-19, atherosclerosis, cancer

## Abstract

There is accumulating evidence that platelets play roles beyond their traditional functions in thrombosis and hemostasis, e.g., in inflammatory processes, infection and cancer, and that they interact, stimulate and regulate cells of the innate immune system such as neutrophils, monocytes and macrophages. In this review, we will focus on platelet activation in hemostatic and inflammatory processes, as well as platelet interactions with neutrophils and monocytes/macrophages. We take a closer look at the contributions of major platelet receptors GPIb, α_IIb_β_3_, TLT-1, CLEC-2 and Toll-like receptors (TLRs) as well as secretions from platelet granules on platelet–neutrophil aggregate and neutrophil extracellular trap (NET) formation in atherosclerosis, transfusion-related acute lung injury (TRALI) and COVID-19. Further, we will address platelet–monocyte and macrophage interactions during cancer metastasis, infection, sepsis and platelet clearance.

## 1. Platelets

Platelets are key drivers of hemostasis and thrombosis. Dysfunction in platelet adhesion, production and clearance lead to cardiovascular diseases (CVD). According to the World Health Organization (WHO), CVDs are the leading cause of death worldwide, with an estimated 17.9 million deaths in 2021 [1]. After vessel injury, platelets adhere to the endothelial matrix and become activated, which leads to thrombus formation and restoration of vascular integrity. In recent years, there has been accumulating evidence that platelets play roles beyond thrombosis and hemostasis, e.g., in inflammatory processes, infection and cancer [2]. In this review, we will focus on the role of platelet activation in hemostatic and inflammatory processes, as well as their interactions with innate immune cells (neutrophils and macrophages) in disease.

### 1.1. Platelet Adhesion

After being released from megakaryocytes in the bone marrow, platelets circulate for 7–10 days in the vasculature before they are cleared, mainly by macrophages in the liver [3]. Their adhesion receptors and biomechanical properties make platelets efficient guardians of hemostasis.

Platelets possess three different types of storage granules (α-granules, δ-granules and lysosomes), which contain receptors and soluble proteins as well as bioactive molecules, which play important roles not only in hemostasis, but also during inflammation and immunity [4]. Hydrodynamic shear forces generated by the blood flow cause erythrocytes to flow in the center of the vessel and push platelets to the vessel wall. After vascular injury, interactions between adhesion receptors on platelets and extracellular matrix (ECM) proteins exposed by the vascular injury decrease the velocity of platelets and allow thrombus formation.

Under low shear conditions present in the venous circulation, integrin α_2_β_1_, α_5_β_1_ and α_6_β_1_ mediate platelet binding to the ECM proteins collagen, fibronectin and laminin. At higher shear rates present in arteries and arterioles, efficient platelet adhesion depends on the von Willebrand factor (vWF) [5]. The vWF is a large glycoprotein produced by megakaryocytes and endothelial cells that circulates in the vasculature in a complex with coagulation factor VIII and is stored in the α-granules of platelets or Weibel–Palade bodies of endothelial cells [6,7]. A shear-induced conformational change of vWF leads to exposure of the A1, A2 and A3 domains and initiates binding of collagen to the A3 domain [8]. Subsequent binding of A3 to collagen further triggers the exposure of the A1 domain. Under lower shear rates, adhesion is mediated through integrin α_IIb_β_3_ on the surface of platelets binding to immobilized vWF on collagen via its C1 domain [9,10,11]. Under higher shear rates, the adhesion of platelets is mainly mediated by vWF binding to glycoprotein Ib (GPIb). The A1 domain on vWF binds to a leucine-rich repeat on the N-terminal domain of GPIb. After vWF-GPIb binding, the stalk region of GPIα unfolds, tensile stress is generated and downstream signaling is initiated [11]. Intracellular signaling leads to granule exocytosis and the activation of α_IIb_β_3_-integrin. vWF binding is characterized by a rapid on–off binding to α_IIb_β_3_ and GPIb, allowing transient interactions [12,13,14]. A stable connection between platelets and collagen is achieved by collagen–GPVI binding, leading to full platelet activation. Soluble agonists, for instance, fibrinogen, ADP, thromboxane A2 (TxA_2_) and thrombin, further support platelet activation and aggregation [9,15].

### 1.2. GPIb Is a Key Receptor for Platelet Activation and Aggregation

GPIb is exclusively expressed on the surface of platelets and megakaryocytes and is part of the GPIb-IX-V complex, which consists of four type-I transmembrane polypeptides: GPIbα, GPIbβ, GPIX and GPV. GPIb-IX-V transduces signals via interactions with the cytoskeleton and without a catalytic intracellular domain [16]. Using an optical tweezer setup with single-molecule force measurement, Zhang and colleagues discovered a mechanosensitive domain in GPIbα, which might be responsible for shear force sensing [17]. Downstream of GPIb, Src/Syk kinases are activated as well as phospholipase C (PLCγ2). This leads to Ca^2+^ release, phosphoinositide 3-kinase (PI3K) and protein kinase C (PKC) activation [14,18]. Consequently, the binding of the tyrosine kinase Syk leads to the phosphorylation of the linker for activation of T cells (LAT) and the further recruitment of phosphoinositide 3-kinase (PI3K). Phosphatidylinositol (4,5)-bisphosphate (PIP2) is converted by PI3K into phosphatidylinositol (3,4,5)-triphosphate (PIP3). Subsequently, PLCγ2 cleaves PIP3, which generates inositol-1,4,5-triphosphate (IP3) and diacylglycerol (DAG). DAG initiates Ca^2+^ release, which, in turn, activates PKC. This ultimately results in granule secretion and the upregulation of activated integrin with increased ligand affinity to further enhance platelet aggregation [18,19].

In 1985, Harmon and Jamieson found that thrombin can bind to platelets via GPIb [20]. A study by Dörmann, Clemetson and Kehrel showed that, together with protease-activated receptor-1 (PAR-1) activation, platelet–platelet interaction and α_IIb_β_3_-fibrinogen binding, GPIb binding to thrombin is an essential part in platelet pro-coagulation activity [21]. An analysis of the protein crystal structure revealed that thrombin shares a binding site with vWF [22,23]. It was hypothesized that thrombin interferes with vWF–GPIb binding, thereby also contributing to an anti-coagulant activity, but studies providing definitive proof are missing [24].

P-selectin is a transmembrane protein stored in α-granules and Weibel–Palade bodies within platelets and endothelial cells, respectively. The translocation of P-selectin to the surface of activated platelets mediates inflammatory processes by interacting with P-selectin glycoprotein ligand-1 (PSGL-1) on leukocytes and endothelial cells, but also on platelets [25]. Interestingly, PSGL-1 and GPIb share structural similarities. Both are sialomucins with a cluster of O-linked carbohydrates and express a cluster of anionic amino acids with sulfated tyrosine [26]. Studies suggest that P-selectin on platelets binds to PSGL-1 on endothelial cells after vessel destruction and promotes adhesion [27].

Filamin A is an actin-binding protein and is essential in platelet adhesion and aggregation. It modulates vWF–GPIb-IX-V interactions by binding to the cytoplasmic tail of GPIbα and supporting the interaction between GPIb-IX-V and actin [28]. Experiments using platelets from mice with a transgenic GPIbα containing a defective filamin A binding site revealed impaired platelet adhesion and loss of membrane integrity under high shear conditions [29].

Interactions between GPIb and coagulation factors such as factor XI, factor XII, factor IX, factor VII, kininogen, protein C and vitamin K have been described. It is assumed that GPIb localizes factor IX to the platelet surface to support thrombin formation [30]. These factors not only support platelet aggregation but are essential in hemostasis [31].

### 1.3. CLEC-2 in Platelet Activation and Immunity

C-type lectin-like receptor 2 (CLEC-2) is a type-II transmembrane receptor expressed as a dimer on platelets, neutrophils, dendritic cells (DCs) and Kupffer cells with an extracellular lectin-like domain. CLEC-2 contains an immunoreceptor tyrosine-based activation motif (ITAM) with a single YXXL motif (hemITAM) necessary for direct signal transduction through the tyrosine kinase Syk [19,32].

Podoplanin is the main binding partner of CLEC-2, expressed on lymphatic endothelial cells and also on inflammatory macrophages. The snake venom rhodocytin is another ligand for CLEC-2 and is frequently used to study its signaling [33]. CLEC-2–podoplanin binding initiates platelet activation and aggregation similar to GPVI through Src, LAT, PLCγ2 and DAG signaling [34,35]. Experiments with CLEC-2-deficient mice showed defective thrombus formation as well as prolonged tail bleeding time in vivo, indicating that CLEC-2 plays an essential role in hemostasis and thrombosis [36].

CLEC-2 also contributes to thromboinflammation; e.g., podoplanin–CLEC-2 interactions on neutrophils or DCs lead to increased protrusions and motility [34]. CLEC-2 activation leads to DC migration to the lymph node and T cell zone to initiate an immune response [37]. Studies showed that CLEC-2 is also involved in hematogenous tumor metastasis by mediating platelet aggregation at tumor sites. Platelet aggregates protect cancer cells and secrete factors to promote tumor growth and proliferation [38,39].

### 1.4. TLT1

The TREM-like transcript 1 (TLT-1) is part of the triggering receptor expressed on the myeloid cells (TREM) protein family. It has a single Ig superfamily V-set domain and signals downstream via two immunoreceptor tyrosine-based inhibitory motifs (ITIM). During platelet production, TLT-1 is stored in α-granules. After platelet activation, it is translocated to the plasma membrane [40] and also released in soluble form. Soluble TLT-1 (sTLT-1) supports the aggregation of platelets during vascular injury [41] and is also involved in inflammatory diseases and sepsis where sTLT-1 levels are increased [42].

### 1.5. Toll-like Receptors (TLRs) and Thromboinflammation

Through their Toll-like receptors (TLRs), platelets can sense pathogens such as bacteria and viruses. For instance, lipopolysaccharides (LPS) or viral infections can trigger a pro-inflammatory platelet response with the release TNF and IL-1 [43,44]. Platelets express all nine TLRs. Platelets TLR2 and TLR4 play a major role by binding gram-positive and gram-negative bacteria, respectively [43]. TLR4 is expressed as a homodimer, whereas TLR2 forms heterodimers with TLR1 or TLR6. To a lower extent, platelets also express endosomal TLRs (TLR3, TLR7 and TLR9), which allow for the recognition of viruses [2,45]. The activation of platelet TLRs leads to aggregation but also a pro-inflammatory response [46]. TLR1/TLR2 stimulation with a synthetic agonist (Pam3CSK4), for instance, triggers platelet aggregation. Moreover, the secretion of pro-inflammatory cytokines and chemokines such as RANTES (regulated upon activation, normal T cell expressed and presumably secreted), PDGF, PF4 and neutrophil extracellular trap (NET) formation was observed [47]. The TLR4 response to LPS is well established and leads to the release of cytokines, IL-1β synthesis, platelet–neutrophil aggregate formation and NET formation [45,48]. TLR4 stimulation also potentiates platelet activation in response to hemostatic agonists such as collagen through Akt and ERK signaling [49]. In conclusion, TLRs on platelets have an impact on platelet aggregation but also contribute to thromboinflammation [45].

### 1.6. Platelet Granules

Platelets communicate with their environment not only via membrane receptors but also via soluble factors released from granules, which also contain receptors that are recruited to the platelet surface upon stimulation. Human platelets contain around 40–80 α-granules, 4–8 dense granules and a few lysosomes [42]. α-granules contain more than 300 different proteins, whereas dense granules contain bioactive amines and nucleotides, such as ADP and ATP, as well as magnesium and calcium (Table 1). Granule development starts during the maturation of platelets by megakaryocytes within the bone marrow [50,51].

The packaging of α-granules takes place after protein synthesis in megakaryocytes but also mature platelets in the multivesicular bodies that are derived from the *trans*-Golgi network [50]. In addition, platelets are also able to pick up plasma proteins and pack them into granules by endocytosis [54]. These processes are mediated by adapter proteins (AP1, AP2 and AP3), soluble N-ethylmaleimide-sensitive factor (NSF) attachment protein receptor (SNARE) proteins and regulators, clathrin molecules, biogenesis of lysosome-related organelles complex (BLOC-1 and BLOC-2) as well as vacuolar protein sorting-associated proteins (VPS33B and VPS16B). A summary of major proteins involved in packaging as well as the factors stored in α-granules can be found in Table 1, as discussing all of them would go beyond the scope of this review [42,50,51,55,56,57].

Upon platelet activation and following Src kinase and PKC signaling, dense granules release soluble factors such as calcium, ATP, ADP and 5-hydroxytryptophan (5-HT). These factors directly activate platelets via receptors such as the purinergic receptors P2Y_x_ in an autocrine and paracrine way [2,50,52].

Defects in platelet granules can lead to serious diseases. One example for an α-granule defect is the gray platelet syndrome (GPS). Mutations in NBEAL2 cause GPS, leading to impaired granule packaging in megakaryocytes and the absence of α-granules in platelets. Symptoms range from a reduced platelet count to bruising susceptibility, myelofibrosis and epistaxis [57,58,59]. One example of a defect in dense granules is the Hermansky–Pudlak syndrome. This is caused by the mutation of HPS and CHS proteins, such as the AP-3 and Rab38, and leads to a bleeding disorder, recurrent infections and ceroid disposition [60].

## 2. Platelets and Neutrophils Interactions in Homeostasis and Pathology

Neutrophils are granulocytic cells that are part of the innate immune system and are the first leukocytes to infiltrate the wound site [61]. Interactions between platelets and neutrophils are crucial for inflammation and immunity [61]. Currently, the platelet–neutrophil crosstalk is well characterized (Figure 1), both in physiologic and pathologic conditions, where the thromboinflammatory response is often described as “a vicious cycle” [62]. Upon vessel wall damage, platelets are activated, adhere, change their shape, secrete their granule contents and start to form aggregates [63]. Activated platelets are characterized by P-selectin exposure, which is critical for both hemostasis and platelet–immune cell interactions [64].

### 2.1. Platelet Activation in Thrombi and/or Circulation Can Cause Neutrophil Incorporation

Platelet P-selectin is a ligand for non-activated neutrophil PSGL1 and the P-selectin–PSGL1 bond enables neutrophil attraction to sites of thrombus formation under low shear stress [64]. PSGL1 is constantly associated with the ITAM-receptor FcRɣ, which is phosphorylated by SFK upon PSGL1 association with P-selectin [65,66]. This results in Syk tyrosine kinase activation, which phosphorylates ADAP adaptor protein and leads to the initiation of PLCɣ2-mediated IP3 production and calcium signaling [65]. On the other hand, ADAP phosphorylation by Syk also results in PI3K activation, which produces PIP3 [67]. PIP3 and calcium synergistically activate the small GTPase Rap1 by inhibiting Rap1 GAP and activating Rap1 GEF, correspondingly [68]. Rap1 activation is the key step in the “inside-out” activation of neutrophil β2-integrins Mac1 and LFA1 [66]. Mac1 and LFA1 then reinforce neutrophil attachment to platelets upon interaction with JAM3 [69], ICAM-2 [70], GPIbɑ [71] and fibrinogen, bound to platelet ɑ_IIb_β_3_ integrin [72]. Recently, a new player in platelet–neutrophil interactions, SLC44A2, was identified. This protein is found on neutrophils and can bind to platelet ɑ_IIb_β_3_ integrins under flow [73]. Neutrophil integrin activation, ligation and clustering results in “outside-in” signaling, possibly mediated by Syk and SFK tyrosine kinases, which enhances neutrophil activation [65,66].

The capability of platelets to induce neutrophil activation is not limited to P-selectin–PSGL1 interaction. Platelet ɑ-granules contain a set of chemokines, which are potent stimulators of neutrophil activity. Among the most abundant are CXCL4 (PF4) [74] and CXCL7 (NAP2) [75]. PF4 has been shown to mediate neutrophil Mac1 and LFA1 integrin activation [76,77]. However, no direct evidence for the presence of a PF4 receptor on the neutrophil surface has been presented to date. It has been proposed that chondroitin-sulfate proteoglycan could serve as a PF4 receptor; final proof, however, remains lacking [78]. On the contrary, mechanisms of NAP-2-induced neutrophil activation are well understood. NAP-2 is produced from its precursor β-thromboglobulin upon release by neutrophil cathepsin-G [79]. NAP-2 acts through CXCR1 and CXCR2—GPCR receptors which regulate both neutrophil activation and chemotaxis [80,81]. It is noteworthy that NAP-2 can cause CXCR2 internalization upon activation, which attenuates neutrophil responses [82]. In addition to NAP-2, platelets also contain CXCL1, which similarly activates CXCR1 and CXCR2 [83]. Both CXCR1 and CXCR2 cause the G_βɣ_-dependent initiation of PLCβ activation and further calcium signaling [84,85]. Interestingly, chemokines can form heterodimers and, thus, can enhance and modulate the activity of each other [86]. For example, PF4 can form pairs with RANTES (CCL5) [87]. Whether such pairing significantly affects platelet–neutrophil interactions has yet to be fully elucidated. Platelets also secrete HMGB1, which can activate both RAGE and TLR4 receptors on neutrophils and, thus, initiate NET formation [88,89]. Finally, upon platelet δ-granule secretion, ADP and ATP are released, which can activate neutrophil purinergic receptors [90]. Other molecules which can mediate neutrophil activation upon platelet granule secretion include TGFβ, CD40L and IL-1β [91]. However, these substances appear to be more potent activators of monocytes instead of neutrophils and their role in platelet–neutrophil interactions is minor [91].

### 2.2. Immunothrombosis Pathways

Not only platelets can be the initiators of platelet–neutrophil association. Inflammation can cause endothelial dysfunction due to high levels of pro-inflammatory cytokines (IL-1, IL-6 and TNF) as well as ferritin [92]. This results in the activation of endothelial cells and expression of P-selectin [92,93]. Neutrophils interact with endothelial P-selectin and become activated in the same manner as upon interaction with platelet P-selectin [65,66]. On the other hand, bacterial infection can activate neutrophils via PAMP/DAMP receptors TLR2, TLR4, Siglec-14, RAGE and CR3 [94]. Neutrophils can also trap viruses into their endosomes. This leads to the decapsulation of the virus and neutrophil TLR7 and TLR8 receptor activation [94]. Activated neutrophils secrete contents of their azurophilic granules, including cathepsin G [95] and NE [96]. Cathepsin G can activate platelet PAR1 receptor in a thrombin-like manner [97]. The same has been assumed for NE; however, no direct experimental evidence is available [91].

Upon hetero-aggregate formation, platelets and neutrophils can activate each other reciprocally. One of the most elegant pathways of such mutual influence is the transcellular synthesis of eicosanoids. Arachidonic acid (AA) is a component of membrane phospholipids, which is released by cytosolic phospholipase 2 (PLA2) upon cell activation [98]. In platelets, AA metabolism by COX1 results in the formation of thromboxane A2 (TxA2)—an essential secondary mediator of platelet activation [63]. However, upon AA release from platelet phospholipids, it can be transferred to neutrophils [91]. Neutrophils are abundant in 5-lipoxygenase (5LO), which is an integral nuclear membrane protein. 5LO activity results in the transition of AA into leukotriene A4 (LTA4), which is further metabolized into leukotriene B4 (LTB4) [91,99]. Alternatively, it can be metabolized into leukotriene C4 (LTC4). Both LTB4 and LTA4 are important pro-inflammatory agents, capable of both activating neutrophils and causing smooth muscle cell contraction and edema [91]. In both activated neutrophils and platelets, reactive oxygen species (ROS) are formed, which enhance the overall activation of both cell types [100]. Additionally, ROS induce NF𝜿B transcription factor activation in neutrophils [101]. NF𝜿B activation can also be triggered by prolonged interaction with platelets via PSGL1 and β2 integrins [102]. This leads to the acquisition of a pro-inflammatory phenotype by neutrophils, characterized by pro-inflammatory cytokine production and, in some cases, tissue factor (TF) surface expression [102,103]. Neutrophils can also activate platelets by releasing antimicrobial cathelicidins via degranulation or as part of NETs. For example, cathelicidin LL-37 and its mouse homologue CRAMP can bind GPVI on the platelet surface, further stimulating platelet and neutrophil activation [104].

In addition to direct platelet activation, activated neutrophils can trigger the activation of the plasma coagulation cascade via TF expression on their surface as well as the secretion of TF-positive microvesicles [92]. This results in the production of thrombin and fibrin generation, which significantly enhances platelet–neutrophil bond formation and mutual activation. It has been suggested that TF-expressing neutrophils and monocytes are key drivers of immunothrombosis [92,103] underlying microvascular thromboinflammation in severe conditions such as ARDS [92,105]. Another distinct feature of platelet–neutrophil interactions in immunothrombotic conditions is the increased potency of neutrophils to form NETs, thus facilitating coagulation and forming a positive feedback loop.

### 2.3. Platelet–Neutrophil Interactions Result in NETosis

Among the most fascinating features of neutrophils is their capability to release neutrophil extracellular traps (NETs)—cobweb-like chromatin structures, whose prime function is to entangle pathogens (specifically gram-negative bacteria) and to remove them from the circulation [94]. NETosis can be initiated upon neutrophil interaction with fungi, bacteria, immune complexes or activated platelets [94]. The role of platelets in NETosis as well as the overall role of platelets in immunity has emerged only in recent years. Platelet P-selectin interaction with neutrophil PSGL-1 not only results in neutrophil integrin activation by PKC-dependent signaling in neutrophils [66,94]; activated platelets also release HMGB1—which activates the neutrophil RAGE receptors [88]. In addition to HMGB1, activated platelets also secrete the C3a component of the complement system, which, in turn, activates neutrophil C3a receptor [106]. Combined, these stimuli significantly enhance PSGL-1-induced neutrophil activation and can result in the generation of ROS due to NADP oxidase activity in neutrophils [89,94]. ROS stimulate the myeloperoxidase (MPO)-mediated transition of NE from azurophilic granules to the nucleus [94]. NE synergizes with MPO to initiate chromatin decondensation. The next step in NET formation is chromatin histone citrullination through PAD4 [107]. PAD4 activation depends on ROS [108] as well as the cytosolic calcium concentration [109]. PAD4 is also activated downstream of PKC [110,111,112]. The type of NETosis inducer defines the histone citrullination pattern [110,112]. Thus, it can be speculated that different isoforms of PKC are activated downstream of different activators and differentially regulate PAD4 activation. As a consequence, citrullinated neutrophil chromatin, especially H3-histones, are secreted from neutrophils in complex with MPO and NE, and a NET is formed [94]. Citrullinated H3 histones, MPO and NE triple staining have been accepted as the most reliable markers of ongoing NETosis [113].

NETs are potent inducers of thrombus formation, serving as a scaffold for platelet and coagulation factor binding and activation [114]. It has been shown that platelets directly interact with NETs via several ways. Platelets can be activated by H4 histones both in vitro and in vivo [115,116]. Furthermore, citrullinated H3 histones, one of the key markers of ongoing NETosis, have also been shown to activate platelets. Moreover, platelets can interact with C3b attached to NETs [117]. In addition to NE and MPO, NETs also contain cathepsin G. On the other hand, because of their negative charge, NETs can also activate FXII and, thus, activate the contact pathway of coagulation [118]. In line with this, FXII on NETs is able to activate kallikrein [119]. These events manifest in thrombin generation and fibrin deposition among NETs, which attract new platelets from circulation. Finally, NE and other NET-associated proteins such as cathepsin G may also contribute to coagulation by degrading TFPI, which is the key negative regulator of blood coagulation [114,120]. In line with this, NET–fibrin structures are significantly more resistant to plasmin-induced fibrinolysis. Therefore, NETs can either initiate or contribute to the ongoing thrombosis as well as to the tissue damage due to NE and MPO activity [114,121]. It is assumed that the capability of NETs to induce thrombosis is an important mechanism required for pathogen containment. However, excessive NETosis is a potent driver of thromboinflammation and subsequent thromboembolic complications [92].

### 2.4. Endothelium

An analysis of the platelet–neutrophil interactions cannot be complete without mentioning the contribution of endothelial cells. Indeed, blood vessel endothelial cells orchestrate the activation of neutrophils, platelets and the coagulation cascade. Upon activation, endothelial cells secrete the content of their Weibel–Palade bodies including vWF [122]. vWF interacts with platelet GPIbɑ, which also enables the interaction of platelets with neutrophil β2 integrins [66,91], and it has been proposed that vWF potentiates this interaction. vWF can attract neutrophils to endothelium independently of platelets, and this process depends on the activity of ADAMTS-13—a metalloproteinase required for the conversion of ultra-large vWF multimers into functional vWF [123]. Other contributions of endothelial cells to platelet–neutrophil heteroaggregate formation are based on their ability to bind pro-inflammatory chemokines to their surface [124]. It has been demonstrated that CCL5 and CXCL4 become immobilized on the endothelium and thereby attract neutrophils and monocytes downstream of the site of ongoing thromboinflammation [87]. Finally, upon neutrophil extravasation from the blood flow, a complex interplay between endothelial cells and platelets maintains blood vessel wall integrity, and this process is mediated by platelet CLEC-2 and GPVI receptors as well as granule release [125,126].

### 2.5. Platelet–Neutrophil Interactions in Pathologic States

Platelet–neutrophil interactions are crucial in host defense and hemostasis [127]. The number of NETs or platelet–neutrophil complexes (PNCs) in blood has been used as a marker of disease severity [128,129,130,131]. Increased amounts of platelet–neutrophil complexes have been observed in autoimmune diseases [74,132,133], acute and chronic inflammatory conditions [134,135] and cardiovascular diseases [136,137].

Ongoing systemic inflammation in immunological diseases may result in increased platelet–neutrophil interactions and a “vicious cycle” of thromboinflammatory responses [134]. For example, patients with ulcerative colitis show a significantly higher number of PNCs and percentage of activated platelets [128]. Platelet activation was proposed to be associated with excessive NET formation in antineutrophil cytoplasmic antibody-associated vasculitis via TLR-CXCL4 signaling [74]. In systemic sclerosis, platelets interact with neutrophils in vitro and in a mouse model and promote neutrophil autophagy via the HMGB1 pathway [132]. Systemic lupus erythematosus (SLE), the most common type of lupus, is a systemic autoimmune disease characterized by a wide range of autoantibodies and clinical manifestations [133]. It was shown that neutrophil-to-lymphocyte and platelet-to-lymphocyte ratios are significantly increased in case of SLE nephritis as well as neutrophil and platelet size and number. These indicate neutrophil and platelet involvement in the disease progress as well as platelet pre-activation and/or enhanced production [138,139]. NETs were also shown to mediate pathology in SLE and antiphospholipid syndrome [140].

Thromboinflammation is also observed in acute inflammation. Early acute myocardial infarction (AMI) and unstable angina are associated with intense neutrophil activation, in contrast to systemic inflammatory syndromes (stable angina, giant cell arteritis, acute bone fracture). Furthermore, the highest amounts of degranulated neutrophils (measured by MPO presence in neutrophils) were observed 4 h post AMI [135]. MPO depletion was associated with platelet activation and platelet–neutrophil complex formation, and a similar pattern was observed when activated platelets were injected in mice [135]. In a transfusion-related acute lung injury (TRALI) mouse model, an increase in platelet–neutrophil interactions and NETs formation was shown, with less NETs present upon anti-platelet therapy [141]. Increased amounts of NETs were observed in the vessels (post mortem) and plasma of patients with TRALI and ALI [141]. In cases of severe sepsis, only 2% of patients showed neutrophils with unaltered MPO levels, suggesting widespread neutrophil activation [135]. NETs are formed during sepsis and their levels correlate with the severity of organ damage and mortality [142]. A mouse model of sepsis demonstrated that gram-positive bacteria can induce neutrophil activation, PNC formation and platelet activation and aggregation in the infected host [143].

Cardiovascular diseases go hand-in-hand with platelet–neutrophil interactions. For example, high levels of circulating DNA and chromatin, which are evidence of NETosis, were shown to be associated with severe atherosclerosis [136]. Composite biomarker score, calculated using both platelet activation and NETosis markers, was shown to be a risk predictor of acute myocardial infarction [137]. HMGB1 was proposed to play a major role in NETosis during myocardial infarction [144]. A review of platelet–neutrophil interactions in cardiovascular diseases was recently published, which provides more details of the mechanisms involved in different cardiovascular conditions [145]. The authors highlight the importance of procoagulant platelets in ischemic stroke [146] and discuss P-selectin and CD40 involvement in atherosclerosis [147,148] and platelet–neutrophil interplay in thrombus formation.

A hyperinflammatory response of the body to infection (cytokine storm) and infiltration of the lungs by neutrophils is observed in COVID-19 [105,121]. Increased NETosis in COVID-19 patients was thoroughly analyzed and the relative number of NETs per neutrophil was calculated to exclude the influence of neutropenia [121,149]. Active COVID-19 virions as well as platelet-rich plasma of COVID-19 patients induced NETosis in neutrophils of healthy donors [149,150]. Microthrombosis was observed in the lungs of COVID-19 patients and NETs were often found in blood clots surrounded by platelets [151,152]. However, NETosis markers only marginally correlate with clinical markers of disease progression [153]. Changes in D-dimer, fibrinogen and platelet count in COVID-19 indicate the activation of the coagulation cascade [154]. Platelet–leukocyte complexes were elevated in COVID-19 patients, with even higher numbers observed in ICU patients [155]. Platelets were pre-activated: changes in P-selectin exposure and GPIb shedding and elevated sizes and higher levels of sCD40L and sP-selectin were shown [156,157,158]. A recent review of COVID-19-associated thrombosis, focusing on all potential contributors, including platelets and neutrophils, was published [159]. At present, the exact contribution of neutrophil–platelet interplay to COVID-19 pathogenesis remains to be studied more thoroughly.

## 3. Monocytes

Monocytes are part of the ‘‘mononuclear phagocyte system’’ (MPS), together with macrophages and conventional dendritic cells (cDCs) [160]. Circulating monocytes constitute 10% of the total leukocyte population in humans and 4% in mice [161]. Monocytes play key roles in tissue homeostasis and immunity, representing a versatile and dynamic cell population, composed of multiple subsets, which differ in phenotype, size, morphology and transcriptional profiles [162]. In humans, three different monocyte populations were first identified by morphology and differential expressions of surface markers CD14 and CD16 [163]. The major monocyte subset, accounting for approximately 90% of the total monocyte population, expresses high levels of CD14 and no CD16 (CD14^+^CD16^−^) and are referred to as classical monocytes. The remaining 10% are shared by CD14^+^CD16^+^ intermediate and CD14^low^CD16^+^ ‘‘non-classical’’ monocytes [163,164]. Similarly, in mice, two populations of monocytes can be described based on the expression of Ly6C, CCR2 and CX3CR1. In mice, ‘‘classical’’ monocytes are characterized by the surface marker combination Ly6C^Hi^CX3CR1^low^CCR2^+^ (previously termed inflammatory monocytes), while Ly6C^low^CX3CR1^Hi^CCR2^−^ monocytes are defined as ‘‘non-classical’’ (also termed patrolling monocytes) [165].

Specific functions have been attributed to the different subsets: non-classical monocytes are thought to “patrol” the vessel walls and to have endothelial cell-supporting functions [166,167], while classical monocytes have the capability to cross the endothelium and enter tissues in response to appropriate signals, such as monocyte chemoattractant protein-1 and 3 (MCP-1/MCP-3), contributing to immunological responses [166,168,169,170]. Historically, monocytes were thought to be mainly precursor cells of macrophages and dendritic cells. However, currently, it is appreciated that in steady-state conditions, most of the tissue-resident macrophages undergo self-renewal [171,172] and, for this reason, the monocyte’s role has been reassessed. It has become clear that blood monocytes can develop in several ways, acquiring a particular antigen-presenting capability or maturing into macrophages [173].

Even though these monocytes show a tissue-resident macrophage signature, it appears that they preserve a degree of their monocyte identity, e.g., Ms4a7 expression [174], and respond differently during inflammation [175]. In addition, Ly6C^hi^ monocytes can migrate into tissues and retain their monocyte-like state without differentiation into macrophages. These extravascular Ly6C^hi^ monocytes constitute a local monocyte pool with minimal transcriptional alterations [176]. Monocyte emigration takes place constitutively in steady-state conditions and is increased during inflammation, when they gain pro-inflammatory properties. Under these circumstances, macrophages that develop from circulating monocytes exhibit first a pro-inflammatory and later an anti-inflammatory phenotype [173].

### 3.1. Platelet Interactions with Monocytes/Macrophages

As mentioned before, platelets are equipped with receptors that enable them to sense vessel damage, inflammation or infection to become activated. Once activated, platelets can recruit and interact with leukocytes, including monocytes and macrophages, stimulating mutual activation and the release of cytokines (Figure 2). Direct platelet–monocyte interactions that lead to the formation of heterocellular complexes was discovered decades ago [177]. Upon activation, P-selectin (CD62P) is exposed on the platelet surface, where it can bind to its counterreceptor on myeloid cells PSGL1 [178]. P-selectin/PSGL-1 interactions seem to be the first step in platelet–monocyte aggregation. In support of this, a study demonstrated that the conjugation between platelets and monocytes is abolished by blocking P-selectin [179]. This first association increases the expression of membrane-activated complex 1 (Mac-1) (CD11b/CD18 integrin α_M_β_2_) on the monocyte surface [180], further supporting their interactions with platelets, e.g., through the platelet receptor GPIb, thanks to its I domain, which is homologous to the vWF A1 domain [181] and JAM-C [69]. Platelets and monocytes also come in contact through CD40L–CD40 [182] interactions and monocyte-triggering receptors expressed on myeloid cell 1 (TREM-1) to TLT-1 on platelets [183,184]. Upon activation, platelets adhere to monocytes through bridging molecules such as fibrinogen and thrombospondin [185,186]. Fibrinogen, in this context, has been reported to assist with complex formation, linking Mac-1 on the monocyte surface and integrin αIIbβ3 on platelets [185,187]. In addition to direct interactions, platelet-derived chemokines influence monocyte recruitment and endothelial adhesion and behavior. Platelet CCL5 (RANTES) deposition on inflamed endothelium is relevant for monocyte recruitment [188]. It was also shown experimentally that platelets release large amounts of CXCL4 (PF4) [51,189]. This cytokine enhances monocyte phagocytosis and triggers respiratory bursts [190] via the activation of phosphoinositol-3-kinase PI3K, Syk and p38 MAP kinase [191]. Moreover, platelet CXCL4 induces extracellular signal kinase 1 and 2 (ERK1/2) phosphorylation, which mediates monocyte survival and differentiation as well as Janus kinase (JNK) signaling, which leads to the production and release of cytokines and chemokines [191]. It was also observed that PF4, in collaboration with IL-4, provokes a rapid differentiation of monocytes into specialized antigen-presenting cells (APCs) with unique phenotypical and functional characteristics that are different from macrophages or cDCs [192]. These cells preferentially stimulated the proliferation of lymphocytes and lytic natural killer (NK) cell activity, while they induced only moderate cytokine responses. Recently, Mac-1 and proteoglycans were identified to mediate the binding of PF4 to leukocytes [193].

Once deposited, platelet chemokines can form homophilic as well as heterophilic aggregates, resulting in the further stimulation of their biological activity. For example, RANTES can increase PF4 binding to the monocyte surface, while PF4 drastically enhances RANTES-induced monocyte arrest on endothelial cells [87], predominantly mediated by CCR1 [194]. Platelets also associate with neutrophils to promote monocyte recruitment: platelet CCL5 can form heterodimers with neutrophil HNP1 (alpha-defensin), stimulating monocyte adhesion through CCR5 [195]. Alard and colleagues demonstrated that the disruption of HNP1–CCL5 interactions attenuated monocyte and macrophage recruitment in a mouse model of myocardial infarction.

Activated platelets were also observed to stimulate monocyte chemotactic protein-1 (MCP-1) secretion and intercellular adhesion molecule-1 (ICAM-1) surface expression on endothelial cells by activating the NF-κB pathway, thereby indirectly causing monocyte recruitment [196]. Thus, platelet adhesion to the endothelium and chemokines secreted by platelets greatly contributes to subsequent monocyte adhesion to the vascular wall.

So far, we have provided a general overview of the events resulting in platelet monocyte aggregation, showing how this interaction can lead to changes in cell markers and phenotype and induce mutual cell activation and cytokine production. However, what are the roles of these aggregates? Multiple studies have investigated the mechanism and impact of platelet–monocyte aggregates in a variety of experimental and pathological contexts. Elevated numbers of circulating blood monocyte–platelet complexes were proposed to play a role in the pathogenesis of acute coronary syndromes and atherosclerosis [197,198] and were detected in different diseases, including rheumatoid arthritis and systemic lupus erythematosus [199], type 1 diabetes [200] and end-stage renal disease [201] suggesting an important role in the pathogenesis of inflammatory diseases.

### 3.2. Platelet–Monocyte/Macrophages Interaction in Atherosclerosis

Atherosclerosis is a chronic inflammatory disease, comprising an intricated vascular injury, which represents the initial pathological event for several cardiovascular diseases [202]. In this inflammatory environment, circulating lipoproteins, mainly low-density lipoproteins (LDL), become oxidized (oxLDL). Circulating monocytes are recruited to the atherosclerotic lesion site through P-selectin and E-selectin, intracellular adhesion molecule 1 (ICAM-1) and vascular cell adhesion molecule 1 (VCAM-1). Monocytes migrate into the subendothelial space under the influence of chemoattractant molecules, where they differentiate into macrophages in response to locally produced factors such as monocyte colony-stimulating factor (M-CSF) [203]. This differentiation includes the upregulation of scavenger receptor A (SR-A) and CD36, which can mediate the uptake of oxLDL [204]. Once macrophages start to phagocytose oxLDL and minimally modified LDL (mmLDL), they transform into foam cells. Foam cells support smooth muscle proliferation, angiogenesis and plaque formation [205]. Finally, the rupture or erosion of vulnerable atherosclerotic plaques leads to arterial thromboembolic events, which may lead to skeletal muscle ischemia or fatal outcomes such as myocardial infarction and stroke [63].

Monocytes and platelets are key players in atherosclerosis [206]. Platelets are considered important drivers of pathogenesis and disease progression thanks to their ability to interact with immune and endothelial cells and through the uptake of LDL [207]. At the site of atherosclerotic plaque formation, platelet activation can be triggered by DAMPs such as oxLDL [208] or podoplanin [209] upregulated on inflammatory macrophages, Th17 T cells and fibroblasts [34]. Specifically, oxLDL has been observed to mediate platelet activation via several pathways, comprising scavenger receptors and platelet-activating factor receptor (PAFR) [208,210]. Platelet PAFR activation by oxLDL occurs simultaneously with classical agonists, ADP and thrombin for example, initiating prothrombotic responses [208]. It has been observed that platelet adhesion to the endothelium precedes leukocyte recruitment and that blocking platelet adhesion with anti-GPIb antibodies or through the genetic deletion of P-selectin reduces leukocyte recruitment to atherosclerotic lesions and plaque development [211,212].

CD40L-CD40 mediated platelet–monocyte interactions facilitate monocyte arrest on inflamed endothelium, accelerating atherosclerosis [147,213]. Platelets support the migration and activation of monocytes, dendritic cells and neutrophils through the release of soluble inflammatory mediators such as CCL5 and PF4, contributing to the progression of atherosclerosis [214]. CXCL4 promotes phenotypic changes in macrophages, resulting in a proatherogenic state with increased susceptibility to foam cell formation [215,216]. Secondly, PF4 may directly aggravate the atherogenic actions of hypercholesterolemia by promoting the retention of lipoproteins. Sachais and colleagues have recently shown that PF4 facilitates the retention of LDL on cell surfaces by inhibiting its degradation through the LDL receptor [217]. Moreover, PF4 facilitates the esterification and uptake of oxLDL by macrophages contributing to foam cell development [218].

### 3.3. Platelet–Monocyte/Macrophage Interaction during Infectious Diseases

Platelets recognize molecular features of microbes, show bacteriocidic activity and contain immunomodulatory mediators essential for alerting and recruiting cells of the immune system [219]. In the following paragraph, platelet–monocyte/macrophage interactions during viral and bacterial infections will be described with a specific focus on platelets and liver macrophages.

Circulating platelet–monocyte and platelet–lymphocyte aggregates have been reported in increased numbers in HIV-infected subjects [220,221]. Liang and colleagues found that platelet aggregates with CD16^+^ inflammatory monocytes are more frequent in HIV-infected patients and associate with increased levels of sCD163, a marker of monocyte activation [222,223]. Platelet aggregates with monocytes, lymphocytes and neutrophils have also been observed during dengue virus infection [224,225]. In 2014, Hottz and colleagues observed that platelet–monocyte complexes correlate with increased vascular permeability in dengue patients. They detected that dengue-activated platelets were able to reprogram monocyte responses *ex vivo*, inducing the secretion of IL-1β, IL-10 and IL-8. [225]. They also found that the interaction of monocytes with apoptotic platelets mediates IL-10 secretion through phosphatidylserine recognition in platelet–monocyte aggregates. Together, their results demonstrated that activated platelets aggregate with monocytes during dengue infection and signal specific cytokine responses that may contribute to the pathogenesis of dengue. Related to SARS-CoV-2, a study by Hottz and colleagues, performed in 2020, found increased platelet activation and platelet–monocyte aggregates formation in severe COVID-19 patients, compared to those with mild disease. They observed that, in COVID-19 patients admitted to the intensive care unit, platelet–monocyte interaction was strongly associated with tissue factor (TF) expression by monocytes as well as with markers of coagulation exacerbation, such as fibrinogen and D-dimers, and all together correlated with a worse outcome [226].

An increasing number of studies confirms the role of interactions between platelets and monocytes/macrophages in immune responses to bacterial infections. Scull and colleagues demonstrated that the incubation of human monocyte-derived macrophages (hMDMs) and autologous activated platelets results in platelet uptake through scavenger-receptors [227]. Following the co-incubation of LPS-activated hMDM with activated autologous platelets, they also observed an increased level of proinflammatory TNF-α, IL-6 and IL-23 cytokines, suggesting that the presence of activated platelets at sites of inflammation may intensify pro-inflammatory macrophage activation. Controversially, other in vitro studies showed the inhibition of pro-inflammatory cytokines during the co-culture of platelets with monocytes–macrophages following bacterial stimulation [228,229,230]. In agreement with this, Hasselbalch and Nielsen observed that activated platelets enhance IL-10 and inhibit TNF-α release from monocytes in a CD40L-dependent manner upon the stimulation of peripheral blood mononuclear cells (PBMCs) with tetanus toxoid (TT), human thyroglobulin (TG), *Escherichia coli* LPS or *Porphyromonas gingivalis* [228]. In vivo studies also showed inconclusive results: studies reported that the formation of circulating platelet–monocyte aggregates could amplify the inflammatory response and predict mortality in older septic patients [231,232]. However, others have demonstrated that platelets contribute to bacterial clearance and resolution of inflammation by regulating macrophage responses [229,233,234,235]. Xiang and colleagues used a murine model of septic shock and showed that platelets inhibit the release of macrophage-derived pro-inflammatory TNF-α and IL-6 and reduce mortality via the COX-1-PGE2-EP4 pathway [229]. Carestia and colleagues revealed that the co-culture of human platelets with monocytes stimulated with LPS reduced not only pro-inflammatory TNF-α and IL-6, but also anti-inflammatory IL-10 release [230]. This was also associated with platelet sequestration of monocyte-derived cytokines. In addition, they evaluated whether platelets could modulate the polarization of monocyte-derived macrophages (M-DMs) and found that monocytes incubated with platelets were predominantly polarized towards the M1 phenotype in a cell-contact-dependent manner by the GPIb-CD11b axis. Lastly, they showed that platelet transfusion in septic mice models, at early time points after *E. coli* infection, increased survival, possibly due to an increase in iNOS+ macrophages, which contribute to efficient bacterial clearance. In summary, although some studies provide information on platelet–monocyte aggregation, more investigation is necessary to gain a better understanding of the biological mechanisms, importance and effects of platelet–monocyte aggregates formed during viral or bacterial infection and sepsis.

The liver is the largest organ in the body, able to filter one-third of the body’s total blood volume/minute with important roles in metabolism, detoxification and defense against bloodstream pathogens [236]. Kupffer cells, discovered by Karl Wilhelm von Kupffer in 1876, are liver-resident macrophages, positioned within the lumen of liver sinusoids. Kupffer cells represent the first line of defense in the circulation and constantly survey the blood to phagocytose and remove pathogens [237,238]. Lee and colleagues showed that Kupffer cells can trap intravascular *Borrelia burgdorferi* [239]. Moreover, blood-borne methicillin-resistant *Staphylococcus aureus* (MRSA) can be captured and killed by Kupffer cells through an interaction of the complement receptor of the immunoglobulin superfamily (CRIg) with Lipoteichoic acid (LTA), a gram-positive bacteria cell wall polymer [240]. However, a minority of *Staphylococci* survive and eventually start to proliferate inside the Kupffer cells [241]. Wong and colleagues discovered that cooperation between platelets and liver macrophages is needed to eradicate blood-borne infections with *Bacillus cereus* and MRSA [242]. Particularly, they observed through intravital spinning-disk confocal microscopy that platelets regularly perform “touch-and-go” interactions under basal conditions. This binding was shown to be mediated by platelet-adhesion receptor GPIb and vWF constitutively expressed on Kupffer cells. Moreover, throughout the infection, bacteria were caught by Kupffer cells and platelets promptly formed aggregates around bacteria on Kupffer cells’ surfaces to contain them in an integrin α_IIb_β_3_- and complement-dependent manner. Using intravital microscopy of the bloodstream of mice infected with *Listeria monocytogenes*, another group showed that bacterial clearance is not a uniform process but a “dual-track” mechanism consisting of parallel “fast” and “slow” pathways. They suggested that slow bacterial clearance is controlled through bacterial opsonization, platelet GPIb binding and the capture of bacteria–platelet complexes via CRIg. The fast clearance of free bacteria, instead, requires Kupffer cell scavenger receptors and is independent of complement and platelets [243,244].

### 3.4. Platelet–Leukocyte Interplay in Cancer Development and Metastasis

Cancer cells can invade nearby tissues and spread throughout the body, resulting in metastasis [245,246]. The tumor microenvironment provides the necessary milieu and blood supply to tumor cells, profoundly influencing the interactions of cancer cells with their surroundings, ultimately determining the fate of the primary tumor: eradication, metastasis or the establishment of dormant micrometastases [247].

Cancer patients frequently present with a hypercoagulable state with an elevated risk for thromboembolic events [248,249]. Tumor cells have been shown to interact with platelets manipulating their physiology, while, on the other side, activated platelets seem to support tumor growth and invasion [250,251]. In particular, tumor cells can activate platelets through the secretion of factors such as ADP [252] or via direct interaction. Platelet ITAM receptors GPVI and CLEC-2, which bind podoplanin expressed at the invasive front of many tumors, participate in cancer pathogenesis [253,254]. Platelet aggregation and secretion of PDGF and TGF-β through CLEC-2 activation [35,255,256] supports tumor cell proliferation and epithelial–mesenchymal transition. High-mobility group box 1 (HMGB1), released by dying tumor cells, on the other hand, interacts with TLR4 on platelets and mediates platelet–tumor cell interaction, promoting metastasis [257]. Through these interactions, platelets become activated and secrete factors that modulate the tumor microenvironment. Vascular endothelial growth factor (VEGF), platelet-derived growth factor (PDGF) and transforming growth factor (TGF) are released from α-granules of activated platelets and stimulate tumor angiogenesis [255,258,259].

Pavlović and colleagues analyzed the contribution of platelets to the progression of hepatocellular carcinoma (HCC) [256]. HCC is the second leading cause of cancer-related death worldwide [260,261]. In more than 90% of cases, this cancer follows an underlying chronic inflammation, fibrosis or cirrhosis [261,262]. Upon liver damage, stellate cells can adopt an activated myofibroblast-like phenotype [262]. This results in an increased deposition of ECM proteins and a release of cytokines, growth factors and products of oxidative stress and, later on, cancer progression [263]. Platelets are an important source of PDGF-β and TGF-β, well-described activators of hepatic stellate cells, with key roles in pro-fibrotic signaling [264].The authors showed that activated platelets support HCC progression in vivo by stimulating tumor cell proliferation and modulating the surrounding hepatic microenvironment. In addition, they reported that platelets mediate macrophage accumulation in the liver and modulate the hepatic inflammatory cell population phenotype towards a pro-tumoral state. Furthermore, they show that activated platelets induce pro-fibrotic signaling and hepatic stellate cell activation, which is in line with previous data showing that platelet CXCL4 and serotonin increase in fibrosis progression [265].

Metastases remain the primary cause for cancer-related death. Hematogenous metastasis comprises multiple steps, including tumor cell dissemination through the circulation, arrest in the microvasculature and, ultimately, the colonization of distant organs [245,246]. Platelets can help the process of cancer metastasis through multiple mechanisms: they enhance epithelial to mesenchymal transition; moreover, they promote tumor cell survival in the circulation, extravasation and colonization at the secondary site [266,267]. Platelets contain a huge reservoir of adhesion molecules such as selectins, integrins and immunoglobulin superfamily proteins, which are essential for promoting platelet adhesion. Cancer cells take advantage of these molecules on the platelet surface to form platelet–cancer cell conjugates through carbohydrate–protein recognition by P-selectin, protein bridging (e.g., by fibrinogen) between integrins as well as PECAM-1–integrin binding [268,269,270]. Cancer cells interacting with platelets can escape NK and other immune cells in circulation [269,271]. Furthermore, cancer cell-bound platelets increase cancer cell adhesion on endothelium and facilitate cancer metastasis [266,271,272]. Interactions between platelets and tumor cells are necessary also for the recruitment of granulocytes to the metastatic site. Labelle et al. showed that cancer cell-bound platelets secrete CXCL5 and CXCL7. Released CXCL7 synergized with CXCR2 to recruit neutrophils from the bloodstream. With the additional help of granulocytes, cancer cells can form an early metastatic niche to establish metastatic foci [273]. Platelet depletion completely suppressed neutrophil recruitment to the metastatic site, strongly indicating that platelets play a critical role in attracting neutrophils.

Gil-Bernabé and colleagues observed that platelet–cancer cell conjugates also recruit monocytes into the early metastatic niche [274], through chemokines released from platelets such as RANTES [275]. Following this, the recruited monocytes/macrophages improve the extravasation of cancer cells by producing VEGF that increases vessel permeability [276] and enhances the formation of metastatic foci [275,277].

This evidence shows that platelets contribute to cancer progression, metastasis and angiogenesis in multiple ways. Leukocytes take part in the interaction with cancer cells and can promote metastases formation. Future research is needed to further investigate the mechanisms involved in platelet–cancer cell–leukocyte interactions.

### 3.5. Platelet Clearance

Hemostatic balance is achieved by the constant clearance of aged, activated or defective platelets in the liver and spleen and the ongoing production of new platelets by megakaryocytes in the bone marrow. How platelet clearance is linked to platelet production is incompletely understood. Desialylation is a marker for aged platelets and the removal of desialylated platelets is an important mechanism of platelet clearance [278,279,280].

The highly glycosylated GPIb-IX-V complex plays an important role in platelet clearance, as the majority of desialylation occurs on GPIbα [281]. The clustering of GPIbα not only mediates platelet aggregation but also clearance. Studies showed that the Fc-independent antibody-mediated activation of GPIbα leads to rapid clearance through hepatic macrophages [282,283,284]. GPIb clustering is one of the main reasons for the rapid clearance of cold-stored platelets after transfusion [285,286], which has a major impact on storage of platelets for transfusion [287].

#### 3.5.1. Platelets Apoptosis

In addition to desialylation, apoptosis is another mechanism involved in the removal of aged platelets and the maintenance of a healthy platelet population. Proteins of the Bcl2-family (BCL-2, BCL-X_L_ and BCL-W) keep platelets viable, whereas pro-death proteins, such as Bax and Bak, are responsible for mitochondrial damage and subsequent apoptosis [288]. Under healthy conditions, pro-survival proteins inhibit Bax/Bak. Cellular stress activates the BH3-only proteins (Bid, Bim, Bad and Bik), which interrupts pro-survival BCL-2-induced Bak/Bax inhibition [289,290]. Activated Bax/Bak generates pores into the mitochondrial membrane, which triggers cytochrome C release, which, in turn, causes the accumulation of Caspase-3 and Caspase-7, leading to DNA damage and the inhibition of transcription [288,291]. How these apoptotic platelets are removed from circulation, however, remains unclear.

A study by Chen and colleagues discovered that the activation of GPIb and subsequent downstream signaling via PI3K initiates Akt-mediated apoptosis. Additionally, opsonization with anti-GPIb antibodies in immune thrombocytopenia (ITP) patients leads to rapid clearance by macrophages through the Fc-receptor [292,293].

#### 3.5.2. Desialylated Platelets Are Rapidly Cleared by Macrophages in the Liver

The glycan composition on the platelet surface plays an important role in platelet clearance. Sialic acid is usually the terminal glycan residue on N- and O-glycans directly connected to β-galactose (β-gal). GPIb-IX-V accounts for ~80% of sialic acid residues on the platelet surface [286]. Platelets with reduced levels of α2,3-linked sialic acids expose β-gal and are cleared through the Ashwell–Morell receptor (AMR), which is expressed on hepatocytes [294]. The AMR exists as a heterooligomeric receptor with two subunits, asialoglycoprotein receptors 1 and 2 (ASGR1/2) [295,296].

AMR-mediated platelet clearance was investigated in AMR knock-out mice that showed reduced platelet clearance as well as platelets with endogenous sialylation defects using *St3Gal4^−/−^* mice, which lack this specific sialyltransferase. [294,297]. In response to platelet clearance, hepatocytes stimulate platelet production by the release of thrombopoietin (TPO) [280]. After the binding of desialylated platelets to the AMR, TPO mRNA expression is upregulated via JAK2 phosphorylation and STAT3 [298].

Experiments on *GPIbα^−/−^* mice showed a significant but not exclusive contribution of GPIb to platelet clearance through desialylation [299]. Studies with anti-GPIb antibodies showed that downstream signaling of activated GPIb leads to the release of granules and the subsequent surface expression of sialidases such as neuraminidase 1 (Neu1) and 3 (Neu3). These enzymes trigger desialylation of adjacent platelets and ultimately lead to increased platelet clearance [280,283,284]. Bacteria-derived neuraminidase also cause desialylation and thrombocytopenia [300]. Shear-induced activation of GPIb-IX-V triggers desialylation: unfolding of the mechanosensory domain of GPIbα leads to platelet activation and increased expression of Neu1 in vivo [301]. Phagocytic α_M_β_2_ integrin is also partially involved in platelet clearance by recognizing exposed N-Acetlyglucosamin on GPIbα [284,302].

Li and colleagues showed that the recognition of desialylated platelets via the AMR also mediates phagocytosis by Kupffer cells via the C-type lectin receptor CLEC4F in mice [303]. However, CLEC4F is not expressed in humans [304], and it is assumed that other lectins adopt the function of CLEC4F in humans [305]. Indeed, macrophage galactose lectin (MGL), together with AMR, was shown to be involved in Kupffer cell-mediated clearance. Intravital microscopy revealed that desialylated platelets directly bind to Kupffer cells, leading to a rapid clearance. Kupffer cell depletion as well as blocking AMR and MGL resulted in an accumulation of desialylated platelets in circulation, suggesting an essential role of Kupffer cells in platelet clearance [306].

## Figures and Tables

**Figure 1 ijms-23-03868-f001:**
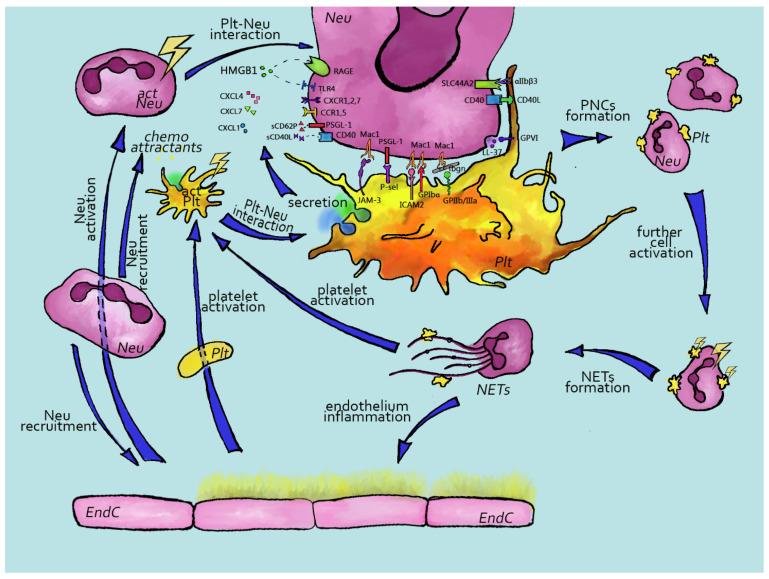
Mechanisms of platelet–neutrophil complex formation. Upon endothelial cell disruption, platelets and neutrophils become activated due to exposure to collagen and activated endothelial cells. Platelet activation results in P-selectin exposure and subsequent neutrophil activation via P-selectin–PSGL1 interaction. Neutrophil activation is further enhanced upon platelet granule content release (CXCL4, CXCL7, CXCL1, HMGB1). The synergistic activation of neutrophils by all these agents via Mac-1, PSGL-1, SLC44A2 and CD40 receptors eventually results in platelet–neutrophil complexes and neutrophil DNA-trap secretion—NETosis. NETs contain the proteases cathepsin G, neutrophil elastase and myeloperoxidase, which can both activate platelets and the coagulation cascade. Multiple positive feedback loops are present at all levels of this system and, thus, both platelet and neutrophils become involved in immunothrombosis and subsequent thromboinflammation. Cell activation is shown by yellow lightnings. Dotted lines represent interactions of soluble mediators and receptors. NET—neutrophil extracellular trap, PNC—platelet–neutrophil complex, Plt—platelet, Neu—neutrophil.

**Figure 2 ijms-23-03868-f002:**
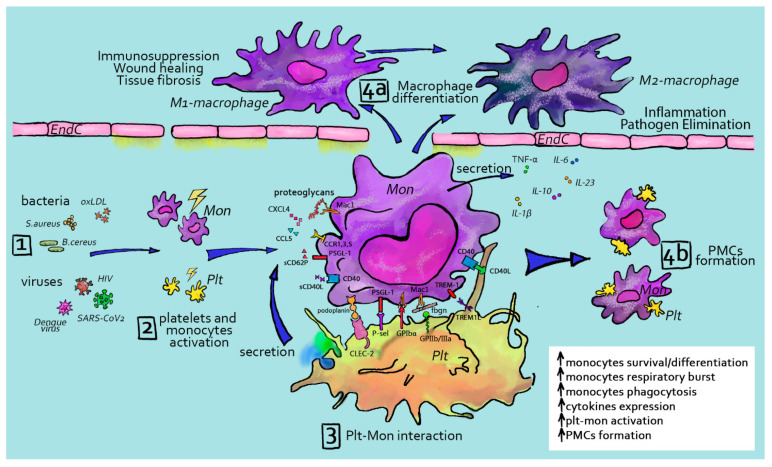
Platelet–monocyte interactions and platelet–monocyte complex (PMC) formation. Several factors such as bacterial or viral infection (**1**) can contribute to the activation of both platelets and monocytes (**2**). Once activated, platelets and monocytes can interact directly through receptors expressed on their surface, and soluble factors released from platelets can further modulate monocyte activity (**3**). This interaction can lead monocytes to extravasate and differentiate towards macrophages (**4a**) or alternatively lead to platelet–monocyte complex formation (**4b**). Cell activation is shown by yellow lightnings. Plt—platelet, Mon—monocyte, PMC—platelet–monocyte complex.

**Table 1 ijms-23-03868-t001:** Properties of α-granules and dense granules [42,50,51,52,53].

	α-Granules	Dense Granules
Structure	Peripheral membrane with an electron-dense nucleoid (chemokines and proteoglycans), an adjacent zone with less electron-dense fibrinogen and a peripheral zone with vWF stored in tubular structures Ovoid 250 nm × 300 nm	Peripherally distributed and electron-dense spherical bodies 150 nm in diameter
Content	**Membrane proteins**: integrins (αIIb, α6, β3), immunoglobulin family receptors (GPVI, Fc receptors, PECAM), leucine-rich family receptors (GPIb-IX-V), tetraspanins, TREM-like transcript-1 (TLT-1), fibrocystin L, CD109, P-selectin and other (CD36, Glut-3) **Soluble proteins**: platelet factor 4 (PF4), fibrinogen, vWF, factor V, factor XI, factor XIII, plasminogen activator inhibitor-1 (PAI-1), α_2_-antiplasmin, chemokines, VEGF, endostatin, FGF, EGF, HGF, IGF, TSP-1, PDWHF, antithrombin, tissue factor inhibitor (TFPI), protein S, protease nexin-2 and pro-inflammatory cytokines	**Soluble proteins**: Bioactive amines (serotonin and histamine), adenine nucleotides (ADP and ATP), Ca^2+^, Mg^2+^ and polyphosphates
Biogenesis	Multi-vesicular bodies (MVBs) derived from the *trans*-Golgi network **Molecules involved**: clathrin, COPII, Adapter proteins (AP-1-3), SNAREs and GTPases	Multi-vesicular bodies (MVBs) derived from the endosomal system **Molecules involved**: Adapter protein 3 (AP-3) and biogenesis of lysosome-related organelles (BLOC 1-3)
Protein sorting	Proteins produced in ER of MKs and sorted via *trans*-Golgi Sorting varies between molecules **Mechanisms involved**: Signal sequence (e.g., Chemokines), glycosaminoglycans (e.g., soluble proteins), aggregation of protein monomers (e.g., vWF), sorting receptors, endocytosis (e.g., plasma proteins such as fibrinogen) and pinocytosis (e.g., plasma proteins such as albumin or immunoglobulin)	Transport of molecules via membrane pumps (e.g., vesicular nucleotide transporter (VNUT) and multidrug resistance-associated protein 4 (MRP4))
Transport	Move along microtubules	Near the plasma membrane
Stimulation and release	Fusing with the plasma membrane by SNAREs (vesicular SNAREs and target SNAREs) Regulated by Sec1/Munc, CDCrel-1, ATPases, SNAPs and Rab proteins	Fusing with the plasma membrane by SNAREs (vesicular SNAREs and target SNAREs) Regulated by Sec1/Munc, CDCrel-1, ATPases, SNAPs and Rab proteins
Function	***adhesion*** and ***coagulation***: fibrinogen, vWF, GPIbα-IX-V, integrin α_IIb_β_3_, GPVI, coagulation factors (V, XI, XIII), precursors of thrombin, prothrombin and kininogen as well as inhibitory proteases (plasminogen activator inhibitor-1 (PAI-1) and α-antiplasmin ***hemostasis***: antithrombin, C1-inhibitor, tissue factor pathway inhibitor (TFPI), protein S, protease nexin-2 and proteinase (plasmin and plasminogen) ***inflammation***: P-selectin, pro-inflammatory factors and chemokines (platelet factor 4 (PF-4), CD40, CD154, CXCL1, CXCL4, CXCL5, CXCL7, CXCL8, CXCL12, CCL2, CCL3 and CCL5) ***antimicrobial host defense***: CXCL4, thymosin-β4, CXCL7, CCL5 and complement factors (C1q, C3 and C4 precursors, C8 and C9) ***angiogenesis***: vascular endothelium growth factor (VEGF), platelet-derived growth factor (PDGF), fibroblast growth factor (FGF), epidermal growth factor (EGF), hepatocyte growth factor (HGF) and insulin-like growth factor (IGF) ***inhibitors of angiogenesis***: thrombospondin-1 (TSP-1) and CXCL-4 ***wound healing***: platelet derived wound healing factor (PDWHF) ***malignancy***: VEGF and adhesive proteins such as P-selectin, vitronectin and fibronectin	Potentiate platelet activation

## Data Availability

Not applicable.

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
