# Peer review of "Beyond Hemostasis: Platelet Innate Immune Interactions and Thromboinflammation"

_ijms, 2022, doi:10.3390/ijms23073868_

Round 1

Reviewer 1 Report

This is a well researched and written review. However it contains only a single diagram  related to a single function of platelets. It would help and improve the  review  if a few more diagrams could be added to demonstrate the complex multifaceted functions of platelets  in immunity and inflammation.  As there is no concluding paragraph to pull all the topics together the diagrams would be helpful,

There are minor English  usage / grammar oversights which should be corrected by the journal  copy editors  before publication.   

Author Response

We thank the reviewer for her/his appreciation of our study. As suggested we have now added an additional figure to summarize platelet interactions with monocytes and macrophages. We have also thoroughly revised the manuscript.

Reviewer 2 Report

The review "Platelet innate immune interactions and thromboinflammation" represents a comprehensive, up-to-date summary in the field of platelet immunity in health and diseases. The review consists of three clear parts covering aspects of platelets aggregation, platelets-neutrophils, and platelets-monocytes interactions. Chapter 1.6 and Table 1 excellently describe differences in composition and function of platelets granules and their functions. It's enjoyable to read and easy to follow and therefore highly recommended for publishing with minor corrections. 

Major comment:

There is detailed visualization of platelets-neutrophil interaction (Fig1), which I found very helpful; I think the readers would appreciate a similar visualization for monocytes-platelets interaction.

Minor comments:

There are multiple punctuation errors in the text; here is just a couple of examples:

Lines 13, 32, 131, …: e.g., instead e.g.

Lines 94: comma before "thereby": "vWF-GPIb binding, thereby also contributing."

Lines 30, 42, 73, ect.: commas before "which"

Lines 152, 171, ect : comma before "whereas"

Author Response

We thank the reviewer for her/his positive comments. As suggested, we have now added a figure on monocyte/macrophage-platelet interactions. We have also thoroughly revised the manuscript text, as suggested.